# Fast Helmet and License Plate Detection Based on Lightweight YOLOv5

**DOI:** 10.3390/s23094335

**Published:** 2023-04-27

**Authors:** Chenyang Wei, Zhao Tan, Qixiang Qing, Rong Zeng, Guilin Wen

**Affiliations:** 1State Key Laboratory of Advanced Design and Manufacture for Vehicle Body, Hunan University, Changsha 410082, China; weicy@hnu.edu.cn (C.W.); tanzhao@hnu.edu.cn (Z.T.);; 2School of Mechanical Engineering, Yanshan University, Qinhuangdao 066004, China

**Keywords:** fast detection, helmet and license plate, lightweight YOLOv5, RHNP dataset, non-truth suppression

## Abstract

The integrated fast detection technology for electric bikes, riders, helmets, and license plates is of great significance for maintaining traffic safety. YOLOv5 is one of the most advanced single-stage object detection algorithms. However, it is difficult to deploy on embedded systems, such as unmanned aerial vehicles (UAV), with limited memory and computing resources because of high computational load and high memory requirements. In this paper, a lightweight YOLOv5 model (SG-YOLOv5) is proposed for the fast detection of the helmet and license plate of electric bikes, by introducing two mechanisms to improve the original YOLOv5. Firstly, the YOLOv5s backbone network and the Neck part are lightened by combining the two lightweight networks, ShuffleNetv2 and GhostNet, included. Secondly, by adopting an Add-based feature fusion method, the number of parameters and the floating-point operations (FLOPs) are effectively reduced. On this basis, a scene-based non-truth suppression method is proposed to eliminate the interference of pedestrian heads and license plates on parked vehicles, and then the license plates of the riders without helmets can be located through the inclusion relation of the target boxes and can be extracted. To verify the performance of the SG-YOLOv5, the experiments are conducted on a homemade RHNP dataset, which contains four categories: rider, helmet, no-helmet, and license plate. The results show that, the SG-YOLOv5 has the same mean average precision (mAP0.5) as the original; the number of model parameters, the FLOPs, and the model file size are reduced by 90.8%, 80.5%, and 88.8%, respectively. Additionally, the number of frames per second (FPS) is 2.7 times higher than that of the original. Therefore, the proposed SG-YOLOv5 can effectively achieve the purpose of lightweight and improve the detection speed while maintaining great detection accuracy.

## 1. Introduction

In recent years, electric bikes have become the ideal means of transportation for the masses due to their convenience, low cost, environmental protection, low carbon, small size, and other characteristics [1]. However, while electric bikes bring convenience to people’s travel, the traffic casualties caused by electric bikes cannot be ignored. According to the latest report on ‘Global Road Safety Status in 2018′ released by the World Health Organization (WHO) [2], about 1.35 million people die in road traffic accidents every year, among which 28% die in accidents with motorcyclists and electric bikes. Relevant research shows that wearing helmets can reduce the death risk of people in above traffic accidents by 42% [3]. Nevertheless, the wearing rate of helmets remains low, especially in some developing countries [2]. In this regard, it is also crucial for the traffic police to check whether the riders are wearing helmets. At present, the inspection of helmet wearing mainly relies on manual labor, which has problems such as involving high law enforcement costs, and being time-consuming and labor-intensive [4].

With the development of artificial intelligence, object detection technology based on deep learning has made remarkable progress in various fields such as road traffic and intelligent security. Combining road monitoring with object detection technology enables automatic detection of whether a cyclist is wearing a helmet. Although methods such as YOLOv4 [5] and YOLOv5 [6] have been successfully applied to helmet detection, they generally cannot meet the memory and real-time requirements when applied in embedded devices with low performance. That is, existing methods still face challenges such as high model complexity and large memory requirements. In addition to helmet detection, rapid identification of license plates is crucial for confirming the identity information of offenders. Only a few studies [7,8,9] have attempted to simultaneously recognize information such as helmets and license plates using a multi-category detection model. However, these methods do not account for the presence of rear passengers, and are susceptible to interference from pedestrians and license plates of parked vehicles, leading to low detection accuracy. To address the aforementioned challenges, this paper aims to develop a rapid detection technology that integrates electric bicycles, riders, helmets, and license plates, achieving rapid and accurate recognition of multi-target objects. This approach is suitable for embedded devices with low performance. The main contributions of this paper include the following:(1)Due to the absence of a public electric bikes, helmet, and license plate detection dataset, the RHNP dataset was created. The dataset includes four categories: the whole of the rider and the electric bike, the head of the person without the helmet, the license plate of the electric bike, and the head of the person wearing the helmet.(2)Since YOLOv5 is difficult to deploy on small mobile embedded devices due to its high computing load and high memory requirements [10], a SG-YOLOv5 model is proposed. Specifically, two lightweight networks, ShuffleNetv2 and GhostNet, were introduced in YOLOv5, and the feature fusion method was changed at the same time. The improved model reduces the number of parameters and computation dramatically, while ensuring the detection accuracy.(3)As for the tedious and time-consuming detection process of violators’ license plates in existing research, this paper draws on and improves the detection method proposed by Allamki L. et al. [9]. Firstly, a scene-based non-truth suppression method is proposed to eliminate the interference of pedestrians’ heads and parked vehicles’ license plates. Then, the violator’s license plate is located through the inclusion relationships of the predicted helmet, rider, and license plate.

The rest of this paper is organized as follows. Section 2 introduces the literature related to this paper. Section 3 introduces the original YOLOv5 network structure. Section 4 introduces the lightweight improvement of YOLOv5, and the application process of the scene-based non-truth suppression method. Section 5 introduces the experiment preparation and setup, and analyses the results of the experiment. Finally, the main conclusions are drawn in Section 6.

## 2. Related Works

Traditional object detection algorithms such as Viola–Jones [11,12], Haar-like [13], and DPM [14] are mainly used in the fields of face detection and pedestrian detection, with slow progress and low performance. Until 2012, the rise of convolutional neural networks (CNN) led to the development of deep learning [15,16,17,18,19], and pushed the field of object detection to a new level. An object detection algorithm based on CNN mainly has two technical development routes: a two-stage detection algorithm and a one-stage detection algorithm [20]. Among them, the two-stage algorithm first generates region proposals, which are preselection boxes that may contain the objects to be inspected, and then classifies samples through CNN. Two-stage detection algorithms include R-CNN [21], SPP-Net [22], Fast R-CNN [23], Faster R-CNN [24], etc. The one-stage algorithm does not use region proposals, but directly extracts features in the network to predict object category and location. Commonly, one-stage detection algorithms include SSD [25], RetinaNet [26], the YOLO series, etc. The YOLO series includes YOLO [27], YOLO9000 [28], YOLOv3 [29], YOLOv4 [5], YOLOv5 [6], etc. In general, the two-stage algorithm has high accuracy but slow speed and cannot meet the requirements of real-time detection, while the one-stage algorithm meets the requirements of real-time detection but its accuracy is slightly lower.

Based on the above-mentioned algorithms, many manufacturers and scholars have carried out research on the detection of electric bicycle helmets and license plates. They either detect the riders wearing helmets [30,31], or both the helmets and the license plates [32,33]. However, few studies have been able to locate and extract license plates of riders not wearing helmets. M. Srilakshmi et al. [7] designed a detection system for the riders without helmets. The system first uses YOLOv2 to detect the rider and motorcycle as a whole in the image and crop out the target box, then uses YOLOv3 to perform helmet detection on the cropped target box, and finally uses YOLOv2 to detect the license plate when the rider is detected not wearing a helmet. Jimit Mistry et al. [8] trained two YOLOv2 models, model 1 was trained from the COCO dataset (80 categories including humans, etc.) and model 2 was trained from the helmet dataset (only one category including helmets). First, model 1 is applied to detect all classes in the images and crop out the human target boxes, then model 2 is used to detect helmets in the cropped images, and finally OpenALPR is used for license plate detection on the cropped images where no helmets are detected. Allamki L. et al. [9] used YOLOv3-tiny to detect five categories in the image, including the helmet, the head of the rider without a helmet, the motorcycle, the rider, and the license plate. Based on the inclusion relation of each target box, first the head of the rider without a helmet is located in relation to the rider, then the rider is located in relation to the motorcycle, and finally the motorcycle is located in relation to the license plate. They used YOLOv3-tiny trained on 11,000 images for 50,000 iterations; the mean average precision (mAP) of the model reached 75%.

It is not difficult to see that for the license plate detection of violators, M. Sirilakshmi [7] and Jimit Mistry [8] et al. have employed multiple object detection models to locate the license plate of violators by cropping the target box, which is tedious and time-consuming. Allamki L. et al. [9] first detected the five categories in the image, and then located the license plate of the violator through the inclusion relationship of the head of the rider without helmet, rider, motorcycle, and license plate of the four target boxes. However, this method did not consider the situation of passengers in the rear seat and the interference of pedestrians and parked vehicle license plates, and the detection precision was not high.

## 3. Yolov5 Network Structure

YOLOv5 is a product of continuous integration and innovation based on YOLOv3 and YOLOv4 [34]. It currently includes four network models, namely YOLOv5s, YOLOv5m, YOLOv5l, and YOLOv5x. Among them, YOLOv5s has the smallest network width and depth [35], has the lowest requirements for mobile hardware, and is more convenient to deploy. Therefore, for this paper we selected the YOLOv5s network model and improved it. The overall structure of YOLOv5s consists of four parts: the image input (Input), the backbone network (Backbone), the Neck, and the output detection head (Head), as shown in Figure 1.

Input adopts Mosaic data augment [36], adaptive image scaling, and adaptive anchor box calculation [37]. As shown in Figure 2, Mosaic data augment randomly uses four images to splice in the way of random scaling, random cropping, and random arrangement, which greatly enriches the samples of the dataset and makes the network more robust. The length and width of the input image may be different, and the original input image is generally uniformly scaled to a certain fixed size and sent to the detection network. To reduce information redundancy, adaptive image scaling can adaptively add the least black edges in the scaled image, so as to accelerate the reasoning speed. The anchor boxes of YOLOv3 and YOLOv4 are pre-defined by K-means, while YOLOv5 uses adaptive anchor box calculation to automatically learn the value of the best anchor box based on the training data during the training process.

The Backbone contains Focus, Conv, C3, and SPP modules [22]. The structure of Focus is shown in Figure 3. First, the input image is downsampled twice for slice operation, then the sliced feature maps are concatenated in the channel dimension, and finally a composite convolution operation is performed after concatenation. The purpose of Focus is to reduce the amount of calculation and speed up the network without losing information. The Conv module is Conv2D+BatchNormal+SiLU, which is used as the basic convolution module in YOLOv5. C3 is the improved version of the CSPDarknet module [38]. Yolov5 contains two different C3 modules, namely C3-True and C3-False. C3-True is used in Backbone for feature extraction of the feature maps, while C3-False is mainly used in Neck to fuse features. Spatial pyramid pooling (SPP) concatenates the feature layer with three multi-scale maximum pooling layers, which can improve the receptive field almost without reducing the speed, so as to solve the alignment problem between the anchor box and the feature layer.

The Neck uses a feature pyramid network (FPN) [39] and a pixel aggregation network (PAN) [40] to construct feature pyramids to fuse the feature layers extracted from the Backbone. FPN transmits and fuses high-level feature information through upsampling, conveying strong semantic features from the top to bottom, while PAN conveys strong positioning features from bottom to top. They fuse the features of the backbone network and the detection layer at the same time, so as to combine the feature information of different scales.

The Head uses anchor boxes of different sizes to predict and classify the feature maps of three different scales, then compares the predicted results with the actual results, and finally generates the loss function for back propagation and optimization parameters. When multiple prediction boxes are generated for a single target in the prediction stage, the optimal prediction box is left by non-maximum suppression (NMS) according to the confidence level, which is the final prediction result.

## 4. Improved YOLOv5s Detection Algorithm

The original YOLOv5s algorithm uses a large number of Conv and C3 structures in the backbone network and feature pyramids, resulting in a large number of parameters and a slow detection speed. In some real application scenarios such as mobile or embedded devices, large and complex models are difficult to apply [10]. To meet the requirements of fast detection of the helmet and license plate, the following improvements have been made to YOLOv5s in this paper: (1) ShuffleNetv2 and GhostNet are introduced to lighten the backbone network and Neck part of YOLOv5s, which reduces the number of model parameters and floating-point operations (*FLOPs*), and increases the detection speed. (2) The feature pyramid feature fusion is replaced with Add, which further reduces the number of model parameters and computation. (3) A non-truth suppression method is proposed to improve the prediction stage and effectively eliminate the interference of non-truth target boxes.

### 4.1. Introducing Two Lightweight Networks: ShuffleNetv2 and GhostNet

#### 4.1.1. ShuffleNetv2 Lightweight Network

The lightweight network ShuffleNetv2 [41] proposed by Questyle summarizes four design principles of the lightweight network through a large number of experiments. The effects of input and output channels, the number of group convolution groups, the degree of network fragmentation, the element-by-element operation on speed and memory access cost (MAC) on different hardware are analyzed in detail:(1)When the number of input and output channels is the same, MAC is the smallest.(2)Group convolution with too large a number of groups will increase MAC.(3)Fragmented operations are not conducive to parallel acceleration.(4)The cost of element-by-element operations (such as ReLU function, Shortcut-add, etc.) cannot be ignored.

According to the above four design principles, ShuffleNetv2 has designed two structures (a basic module and a downsampling module), as shown in Figure 4. Figure 4a is the basic module of ShuffleNetv2, which splits the input feature map into two branches by channel. In the two branches, branch1 selects direct skip connections, and branch2 uses the channel-by-channel convolution with convolution step of 1 and the ordinary 1 × 1 convolution to form depthwise separable convolution, which has the same number of input and output channels for each convolution operation. Finally, the two branches are concatenated, and the channel rearrangement operation is used to enhance the information communication between channels. Figure 4b is the downsampling module of ShuffleNetv2, which removes the channel separation operation compared to the basic module. Branch1 uses the depthwise separable convolution with a convolution step of 2; the convolution step in branch2 is changed to 2.

Generally, 1 × 1 convolution is used before or after the channel-by-channel convolution for two purposes, one is to fuse the information between channels, and the other is to increase or reduce the dimension. In branch2, there is no need to increase or reduce the dimension, only to fuse the information between channels. Therefore, in this paper, the 1 × 1 convolution after the channel-by-channel convolution is removed, which removes a small amount of computational cost. The two structures of the improved ShuffleNetv2 are shown in Figure 5.

Currently, the complexity of a general network model can be measured by both spatial and time complexity. Spatial complexity is the size of the model parameters, which is composed of the weights and bias terms of each convolution kernel in the network. Time complexity (i.e., the amount of computation commonly used) can be measured by floating-point operations (FLOPs); the specific calculation formulae are
(1)$$LOPs=FLOPs_{fc}+FLOPs_{conv}$$
(2)$$FLOPs_{fc}=I\cdot O$$
(3)$$FLOPs_{conv}=C_{in}\cdot C_{out}\cdot K^{2}\cdot H_{out}\cdot W_{out}$$

In the formulae, $FLOPs_{fc}$ and $FLOPs_{conv}$ represent the computational effort of the full connection layer and the convolution layer, respectively; I and O represent the number of input and output nodes of the full connection layer, respectively; $C_{in}$ and $C_{out}$ represent the number of input and output channels for the convolution layer, respectively; K is the size of square convolution kernels, and $H_{out}$ and $W_{out}$ represent the height and width of the output feature map, respectively. Here, since the bias has little effect on the total FLOPs, it is ignored in the calculation. Here, due to the fact that the number of bias terms is much smaller than the number of weights in each convolution kernel, the bias terms have little effect on the total FLOPs and they are ignored in the calculation. From the formulae, the FLOPs of the depthwise separable convolution used by ShuffleNetv2 are:(4)$$FLOPs_{DWS}=C_{in}\cdot K^{2}\cdot H_{out}\cdot W_{out}+C_{in}\cdot C_{out}\cdot H_{out}\cdot W_{out}$$

To minimize MAC, ShuffleNetv2 complies with the first lightweight network design principle, which maintains the same number of input channels $C_{in}$ and output channels $C_{out}$ of the convolution layer in the module. Therefore, the ratio of depthwise separable convolution to the original convolution operation is r:(5)$$r=\frac{FLOPs_{DWS}}{FLOPs_{conv}}=\frac{C_{in}\cdot K^{2}\cdot H_{out}\cdot W_{out}+C_{in}\cdot C_{out}\cdot H_{out}\cdot W_{out}}{C_{in}\cdot C_{out}\cdot K^{2}\cdot H_{out}\cdot W_{out}}=\frac{1}{C_{in}}+\frac{1}{K^{2}}$$

The actual number of input channels is generally large, so the ultimate calculation amount saved is approximately inversely proportional to the square of the convolution kernel size.

#### 4.1.2. GhostNet Lightweight Network

Han K. et al. [42] visualized the output of ResNet50 and found that there were a lot of duplicate and redundant feature maps during the training process. Therefore, a new end-to-side neural network architecture GhostNet is proposed, which provides a Ghost module. The module is designed to generate more feature maps by cheap operation, thus reducing the computational cost.

Ghost, as a plug-and-play module, mainly consists of a small amount of convolution, linear operations, and feature map splicing, as shown in Figure 6. First, a small amount of convolution is used to compress the number of channels in the input image to get feature layer A, and then the feature map of feature layer A is linearly operated one by one to get feature layer B. Finally, feature layer A is identively mapped and spliced with feature layer B to output a new feature layer C. After normal convolution calculation, the output of the input feature map $X_{h*w*c}$ is $Y_{h\prime *w\prime *n}$, expressed as
(6)$$Y_{h\prime *w\prime *n}=X_{h*w*c}\times f_{n*k*k*c}+b,$$
where f is the convolution operation with n convolution kernels of size k*k, the number of channels is c, and b is a convolution bias term. There, h,w,c and h′,w′,n respectively represent the height, width and number of channels of the input and output feature maps, respectively. The amount of computation for normal convolution is n×h′×w′×c×k×k, while the Ghost module convolution is divided into two steps. The first is a small amount of convolution, and the output feature map is $Y\prime_{h\prime *w\prime *m}$; the formula is expressed as
(7)$$Y\prime_{h\prime *w\prime *m}=X_{h*w*c}\times f\prime_{m*k*k*c}+b^{\prime }.\;$$

From the formula, the calculation amount is m×h′×w′×c×k×k. Then, a linear operation is performed on the m feature maps one by one, and each feature map is transformed to generate s feature maps. Since there is an identity mapping at the end, the actual number of transformations is s−1, the size of each linear operation convolution kernel is d×d, and the total calculation amount of linear operations is $m\times \left(s-1\right)\times h\prime \times w\prime \times d\times d$. When m≪n is satisfied, n=m×s is obtained. Thus, the calculation amount of the Ghost module can be expressed as $n/s\times h\prime \times w\prime \times c\times k\times k+n/s\times \left(s-1\right)\times h\prime \times w\prime \times d\times d$. The theoretical acceleration ratio of normal convolution to Ghost module convolution is
(8)$$rate_{s}=\frac{n\cdot h\prime \cdot w\prime \cdot c\cdot k\cdot k}{\frac{n}{s}\cdot h\prime \cdot w\prime \cdot c\cdot k\cdot k+\left(s-1\right)\cdot \frac{n}{s}\cdot h\prime \cdot w\prime \cdot d\cdot d}=\frac{c\cdot k\cdot k}{\frac{1}{s}\cdot c\cdot k\cdot k+\frac{\left(s-1\right)}{s}\cdot d\cdot d}\;.$$

When d=k, the theoretical acceleration ratio is
(9)$$rate_{s}\approx \frac{s\cdot c}{s+c-1}\approx s.\;$$

The theoretical parameter compression ratio of normal convolution to Ghost module convolution is
(10)$$rate_{c}=\frac{n\cdot c\cdot k\cdot k}{\frac{n}{s}\cdot c\cdot k\cdot k+\left(s-1\right)\cdot \frac{n}{s}\cdot d\cdot d}=\frac{c\cdot k\cdot k}{\frac{1}{s}\cdot c\cdot k\cdot k+\frac{\left(s-1\right)}{s}\cdot d\cdot d}.$$

When d=k, the theoretical parameter compression ratio is
(11)$$rate_{c}\approx \frac{s\cdot c}{s+c-1}\approx s.\;$$

From the simplification results, the number of calculations and parameters of normal convolution is about s times that of Ghost module convolution. Thus, using the Ghost module can make the model lighter and faster. Based on the Ghost module, this paper uses the Ghost series modules shown in Figure 7 to replace the Conv and C3 modules in YOLOv5s with lighter modules.

In the figure above, k represents the size of a default convolution kernel, GroupConv represents group convolution, act = false indicates no activation function layer, and DWConv represents depthwise separable convolution.

### 4.2. Add-Based Feature Fusion Method

Convolutional neural networks extract image features by performing convolution operations on input images. Low-level features have higher resolution, accurate target location, and can better reflect the specific content of the image. The features such as contour, edge, color, texture, and shape features are more obvious. After multiple convolution operations, high-level features have stronger semantic information and can better express the image information that humans can understand, but their resolution is lower [39]. The Neck part of YOLOv5 uses FPN and PAN structures to construct a feature pyramid to fuse low-level detail features with high-level semantic features in the Concat-based method. The feature pyramid structure here is designed to increase the resolution of feature maps with smaller resolution but stronger semantics, which can be theoretically fused with the Add-based method.

The two feature fusion methods Concat and Add are shown in Figure 8, where w, h, and m represent the width, height, and number of channels, respectively. It is not difficult to see that the Concat-based method superimposes the feature maps in the channel dimension, which requires that the number of channels for each feature map may be different, but the width and height must be equal. The Add-based method adds elements of corresponding positions, which requires the width, height, and number of channels of each feature map to be equal. Due to the large number of input and output channels, the computation amount of Add is much smaller than that of Concat, so the Concat-based method is replaced with the Add-based method in this paper.

### 4.3. Scene-Based Non-Truth Suppression Method

On real traffic roads, using the trained model to predict images may detect the pedestrian head target boxes and the license plate target boxes of electric bikes parked on the roadside. As irrelevant target boxes, these are interference items for the detection of the helmet and license plate in this paper. At the same time, in order to facilitate the subsequent positioning of the license plate information of riders who do not wear helmets as required, this paper proposes a scene-based non-truth suppression method.

First, four categories in the image are detected: the whole of the rider and the electric bike, the head of the person without the helmet, the license plate of the electric bike, and the head of the person wearing the helmet. Then, non-truth suppression is performed during the prediction stage to remove the interference of the head target boxes of pedestrians and the license plate target boxes of electric bikes parked on the roadside. Finally, only four categories are detected: the whole of the rider and the electric bike, the head of riders without helmets, the head of riders wearing helmets, and the license plate of electric bikes. Figure 9a shows the non-truth suppression process, and Figure 9b shows an example image of the non-truth suppression.

### 4.4. SG-YOLOv5 Network Structure

According to the theoretical analysis of the above chapters, a lightweight YOLOv5 model (SG-YOLOv5) is proposed for the detection of the helmet and license plate of the electric bike. The network structure of SG-YOLOv5 is shown in Table 1.

The ‘From’ column in the table indicates which layer the input of the module comes from, and -1 signals that the input of the module comes from the output of the previous layer. The ‘Params’ column represents the number of module parameters of this layer, the ‘Module’ column represents the module name of each layer, and the ‘Arguments’ column indicates the information of the module parameters, including the number of input channels, the number of output channels, the size of the convolution kernel, and the step. It is worth noting that the second Shuffle_Block downsampling module of SG-YOLOV5 keeps the number of input and output channels unchanged. This operation slows down the surge in the number of subsequent feature layer channels, thereby reducing the model complexity.

SG-YOLOv5 uses the loss function of YOLOv5, which is composed of three parts: the classification loss ($l_{cls}$), the confidence loss ($l_{obj}$), and the position loss ($l_{box}$). $l_{cls}$ and $l_{obj}$ use BCEWithLogitsLoss, and the calculation formulae are as follows:(12)$$Loss=l_{cls}+l_{obj}+l_{box},\;$$
(13)$$l_{cls}=\sum_{i=0}^{s^{2}}I_{ij}^{obj}\sum_{c\in classes}\left(\hat{P_{i}}\left(c\right)log\left(P_{i}\left(c\right)\right)+\left(1-\hat{P_{i}}\left(c\right)\right)log\left(1-P_{i}\left(c\right)\right)\right),$$
(14)$$l_{obj}=\sum_{i=0}^{S^{2}}\sum_{j=0}^{B}I_{ij}^{obj}\left(\hat{C_{i}}log\left(C_{i}\right)+\left(1-\hat{C_{i}}\right)log\left(1-\hat{C_{i}}\right)\right)-\lambda_{{\;}_{noobj}}\sum_{i=0}^{S^{2}}\sum_{j=0}^{B}I_{ij}^{noobj}\left(\hat{C_{i}}log\left(C_{i}\right)+\left(1-\hat{C_{i}}\right)log\left(1-\hat{C_{i}}\right)\right),\;$$
where $S^{2}$ indicates that the image is divided into S×S grid cells, B indicates the bounding box, and $I_{ij}^{obj}$ indicates the *j*-th bounding box of the *i*-th grid cell. If $I_{ij}^{obj}$ contains the target center point, its value is 1, otherwise it is 0, and $I_{ij}^{noobj}$ is the opposite. $P_{i}\left(c\right)$ represents the category prediction probability, $\hat{P_{i}}\left(c\right)$ represents the category true probability, $C_{i}$ represents the confidence value, $\hat{C_{i}}$ represents the intersection of the predicted bounding box and the ground truth box, and $\lambda_{noobj}$ represents the confidence loss coefficient of no target.

The position loss $\left(l_{box}\right)$ adopts CIOU_Loss, which also considers the overlap area, the center point distance, and the aspect ratio between the predicted bounding box and the ground truth box. The specific calculation formulae are:(15)$$l_{box}=1-CIOU=1-\left(IOU-\frac{S^{2}}{c^{2}}-\frac{v^{2}}{\left(1-IOU\right)+v}\right)=1-\left(\frac{A}{B}-\frac{s^{2}}{c^{2}}-\frac{v^{2}}{\left(1-\frac{A}{B}\right)+v}\right)\;$$
(16)$$v=\frac{4}{\pi^{2}}{\left(arctan\frac{w^{gt}}{h^{gt}}-arctan\frac{w^{p}}{h^{p}}\right)}^{2}$$

Among them, A is the area of the intersection of the predicted bounding box and the ground truth box, B is the area of the union of the two bounding boxes, s is the Euclidean distance between the center points of the two bounding boxes, *c* is the diagonal distance of the two bounding boxes, $w^{gt}$ and $h^{gt}$ are the width and height of the ground truth box, respectively, and $w^{p}$ and $h^{p}$ are the width and height of the predicted box, respectively.

## 5. Experiments and Result Analysis

### 5.1. Experiment Preparation and Setup

#### 5.1.1. Dataset Collection

On account of the fact that there was no public dataset available, the experimental dataset came from field shooting, and the shooting locations were multiple traffic roads in Changsha. The relevant indicators and corresponding values of the shooting camera are as follows: S = 1/17 s, EV = 0, F = 1.8, resolution = 3456 × 4608. In order to use license plate detection to obtain the personal information of illegal riders, the shooting angle was the rear and the oblique rear of the electric bikes. Based on this, a total of 2700 pictures were taken to create a dataset, some of which are shown in Figure 10. The labels of the dataset include four categories: rider, no helmet, license plate, helmet. Among them, ‘rider’ refers to the whole of the rider and the electric bike; ’no helmet’ refers to the head of the person without helmet; ‘license plate’ means the license plate of the electric bike; and ‘helmet’ refers to the head of the person wearing the helmet. The dataset is named RHNP dataset (https://drive.google.com/file/d/1zESCktHH3VQfsaDviks7v9inztDQ1IXQ/view?usp=sharing, accessed on 3 March 2023), and randomly divided into train set, validation set, and test set, with a ratio of 7:1:2. As shown in Figure 11, LabelImg software 5.1.1 is used to manually label the collected images. LabelImg software sets the storage format to YOLO format; the label suffix is TXT, and the label and image names are consistent.

The results of visualizing the location distribution of the center points of the target boxes and the size distribution of the target boxes in the dataset re shown in Figure 12. Figure 12a shows the location distribution of the center point coordinates of the target boxes after the size of image resolution is regularized. The darker the color is, the higher the number of center points of the target boxes at this point. In Figure 12b, width and height, respectively, represent the proportion of the width and height of the target boxes to the width and height of the images. It can be seen from the two figures that most of the targets in the dataset are distributed in the lower right, and the proportion of small and medium-sized targets is larger.

#### 5.1.2. Experimental Environment and Parameter Settings

The experimental environment was based on Windows 10, and the deep learning framework used the Pytorch1.7.1 architecture. A NVIDIA GeForce RTX 3080 Ti was used as the video card. The video memory size was 12 GB and the memory size was 64 GB. The specific experimental configuration is shown in Table 2.

In the training phase of SG-YOLOv5, the model was iterated for 500 rounds. To compare it with other models as fairly as possible, the remaining parameter values were consistent with the default settings of YOLOv5. The target box dimensions in the dataset were re-clustered using the K-means algorithm before the model training, the parameters of the initial anchor box were set to [10, 13, 16, 30, 33, 23], [30, 61, 62, 45, 59, 119], [116, 90, 156, 98, 73, 326], the section of image input adopts Mosaic method to enhance the dataset. The input image size was 640 × 640 × 3, the training batch was 16, an SGD optimizer was used for the parameter optimization, the initial learning rate was 0.01, the weight decay coefficient was set to 0.0005, and the learning rate momentum was 0.937. After 3 iterations of the Warmup method with a momentum parameter of 0.8, the learning rate cycle of cosine annealing was entered.

#### 5.1.3. Evaluation Indicators

In object detection tasks, when the intersection over union (IOU) between the prediction box and the truth box is greater than a certain threshold, the prediction result is regarded as a positive sample. Otherwise, it is treated as a negative sample. Based on these analyses, the number of positive samples with correct predictions is counted as TP; the number of positive samples with incorrect predictions is counted as FP; and the number of negative samples with incorrect predictions is counted as FN. In this paper, several evaluation indicators commonly used in the field of object detection were used to evaluate the performance of the algorithm model. The selected indicators mainly included precision (P), recall (R), average precision (AP), mean average precision (mAP), and frames per second (FPS).

Among them, P refers to the correct proportion of all targets predicted by the model; R refers to the correct proportion of all real targets predicted by the model; AP refers to the area under the PR curve (Precision–Recall curve); mAP is the average value of AP of each category, then mAP0.5 represents the mAP when the IOU threshold is 0.5, and mAP0.5:0.95 represents the average mAP at different IOU thresholds (from 0.5 to 0.95, step 0.05). In the above experimental environment, the FPS is obtained by averaging the total detection time of 540 pictures in the test set, and the calculation formulae are:(17)$$P=\frac{TP}{TP+FP}\;,$$
(18)$$R=\frac{TP}{TP+FN}\;,$$
(19)$$AP=\int_{0}^{1}P\left(R\right)\;,$$
(20)$$mAP=\frac{\sum_{i=1}^{n}AP_{i}}{n}\;.$$

### 5.2. Experimental Results

#### 5.2.1. Training Results

The SG-YOLOv5 and YOLOv5s models were trained on the RHNP dataset, using the same experimental environment and parameter settings. After the models converged, the average values of the corresponding evaluation indicators were taken every 50 iterations to obtain the model comparison table shown in Table 3.

It can be seen from the figures that the average precision, precision, and recall of SG-YOLOv5 and YOLOv5s on the RHNP dataset are almost the same. To further confirm the effectiveness of the SG-YOLOv5 model for helmet and license plate detection, the trained SG-YOLOv5 model was tested in a real scene, as shown in Figure 13. It is apparent from the figures that SG-YOLOv5 meets the detection requirements for multiple targets, long-distance small targets, and occluded targets on different traffic roads as well.

#### 5.2.2. Non-Truth Suppression and License Plate Extraction Experiment

This study improves the YOLOv5s prediction stage and proposes a non-truth suppression method. The experimental results are shown in Figure 14. Figure 14a,c are the prediction outputs before improving the prediction stage; Figure 14b,d are the prediction outputs using the non-truth suppression method. A comparison of the figures shows that the non-truth suppression method can effectively remove the interference of pedestrian head target boxes and license plate target boxes of electric bikes parked on the roadside.

The extraction process of offenders’ license plates in this paper is shown in Figure 15. First, the SG-YOLOv5 model was used to predict the image. Then, according to the inclusion relationship of each target box, the no-helmet is located in relation to the rider, and the rider is located in relation to the license plate. Finally, the license plate is extracted. It can be seen that for this study we only needed to train a single object detection model, abandoning the operation of locating violators’ license plates by cropping the target box in the existing researches, and greatly improving the detection efficiency.

#### 5.2.3. Comparative Experiment

To verify the effectiveness of the SG-YOLOv5 model proposed in this paper for the detection of electric bikes, helmets, and license plates, a comparative experiment was conducted on the self-made RHNP dataset. That is, SG-YOLOv5 was compared with several classic one-stage object detection algorithms. Under the same experimental environment, the experimental results obtained on the test set are shown in Table 4. Because the computing power of the high-performance GPU platform was sufficient and the memory bandwidth was large, the lightweight model had little advantage over the original model. However, the lightweight model designed in this paper will mainly be applied to devices with limited memory and computing resources. Therefore, the test environment for FPS values was a personal computer (PC) with limited computing resources, the CPU was an AMD Ryzen 7 4800H, and the GPU a NVIDIA GeForce GTX 1650Ti.

It can be seen from the table that on the test set, the mAP0.5 of SG-YOLOv5 is 0.2% higher than that of YOLOv5s, and the FPS of SG-YOLOv5 is 2.7 times that of YOLOv5s. Meanwhile, the number of parameters is reduced by 90.8%, the FLOPs are reduced by 80.5%, and the model file size is reduced by 88.8%. Compared to the lightweight networks, YOLOv5s-MobileNetv3 and YOLOv5s-GhostNet of the same baseline, the mAP0.5 of SG-YOLOv5 is respectively 3.9% and 0.8% higher. Meanwhile, the number of parameter, FLOPs, FPS and the model size of SG-YOLOv5 are all optimal. Compared to the lightweight network YOLOv5s-ShuffleNetv2, although the FPS and FLOPs of SG-YOLOv5 are not dominant, the mAP0.5 is 1.1% higher and it is superior in the number of parameters and the model file size. Compared to other one-stage object detection algorithms such as YOLOv3, YOLOv3-tiny, YOLOv3-spp, YOLOv4, and SSD, the SG-YOLOv5 proposed in this paper is optimal in terms of various evaluation indicators. Compared to the latest YOLOv7, the mAP0.5 of SG-YOLOv5 is slightly lower, but all other evaluation indicators are significantly improved. Combining the complexity of each model and the actual detection effects, it can be seen in general that SG-YOLOv5 performs better among these models.

## 6. Conclusions

This paper proposes a lightweight model SG-YOLOv5 based on YOLOv5 for detection of electric bikes, helmets, and license plates. Two strategies are used to reduce the number of model parameters and FLOPs. First, the YOLOv5s backbone network and the Neck part are lightweighted by combining ShuffleNetv2 and GhostNet. Second, the feature pyramid feature fusion method is replaced with Add. Compared to YOLOv5s, the mAP0.5 of SG-YOLOv5 is almost the same, but the number of parameters of SG-YOLOv5 is reduced by 90.8%, FLOPs are reduced by 80.5%, the model file size is reduced by 88.8%, and the FPS is 2.7 times of YOLOv5s. At the same time, compared to other lightweight networks, SG-YOLOv5 also has great advantages; for example, the model can be better deployed on mobile terminals or embedded devices with limited memory and computing resources. Further, in order to eliminate the interference of pedestrians’ heads and parked electric bikes’ license plates, this study improves the YOLOv5s prediction stage. That is, a non-truth suppression method is proposed, and this method is applied to SG-YOLOv5.

Furthermore, in order to eliminate the interference of pedestrians’ heads and parked vehicles’ license plates, this paper improves the YOLOv5s prediction stage and proposes a non-truth-value suppression method. Finally, the license plates of riders without helmets are located and extracted according to the inclusion relationship of the target box, so as to determine the identity information of violators.

However, due to the influence of many complex factors in reality, the detection algorithm proposed in this paper still has some shortcomings and deserves further study:

(1)In this paper, the license plate detection of the riders without helmets is realized only in a relatively ideal traffic environment. However, in complex traffic environments, such as crowds of pedestrians and riders in the image, there may be overlapping of heads or license plates, resulting in the unclear positioning of violators and corresponding license plates. Therefore, the corresponding research work should be carried out in more complex traffic environment in the future.(2)This paper only classifies and detects several related categories. Despite the interference of pedestrians and parked electric bikes being excluded, the identification of the license plate information of the violators still needs to be further improved. For example, after locating the license plate of the illegal rider, optical character recognition (OCR) of the license plate should be performed.(3)Foggy backgrounds and image motion blur are not considered in this paper. In future work, we can deploy a defogging algorithm and a deblurring algorithm to clear the image, and then carry out target detection on the image.(4)As for future work, we also hope to develop a license plate identification system for illegal riders, with the complete interface and the software and hardware platform to better protect peoples’ safe travels.

## Figures and Tables

**Figure 1 sensors-23-04335-f001:**
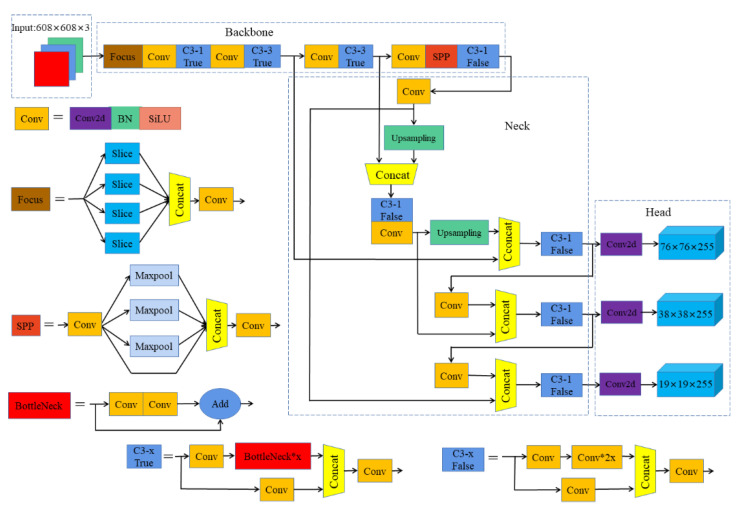
Network structure of YOLOv5s.

**Figure 2 sensors-23-04335-f002:**
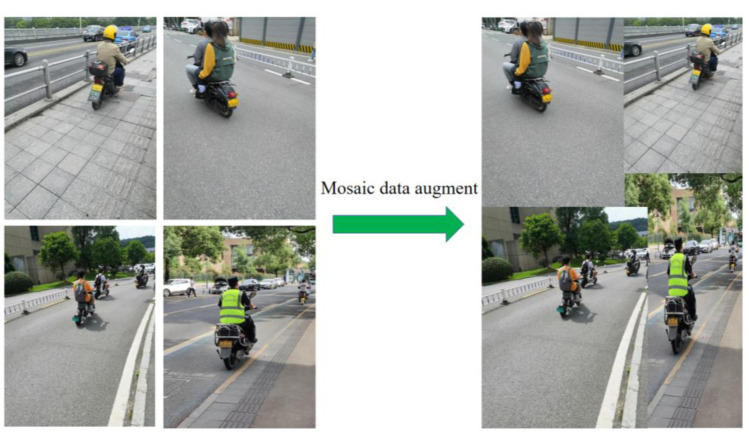
The image example of Mosaic data augment.

**Figure 3 sensors-23-04335-f003:**
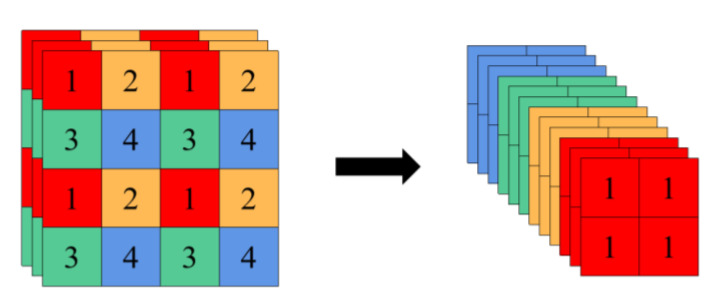
Focus slice operation.

**Figure 4 sensors-23-04335-f004:**
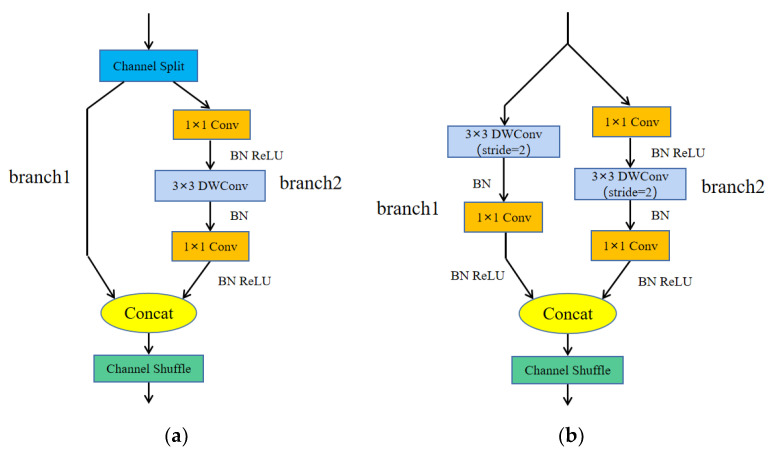
ShuffleNetv2 module. (**a**) Basic module, and (**b**) downsampling module.

**Figure 5 sensors-23-04335-f005:**
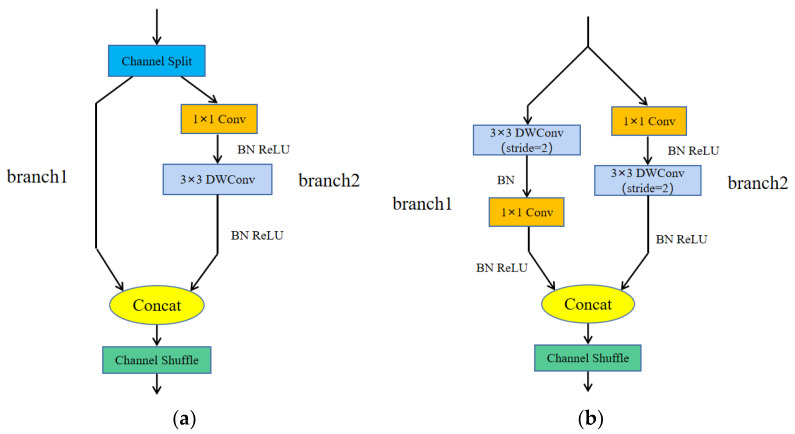
Improved ShuffleNetv2 module. (**a**) Improved basic module, and (**b**) improved downsampling module.

**Figure 6 sensors-23-04335-f006:**
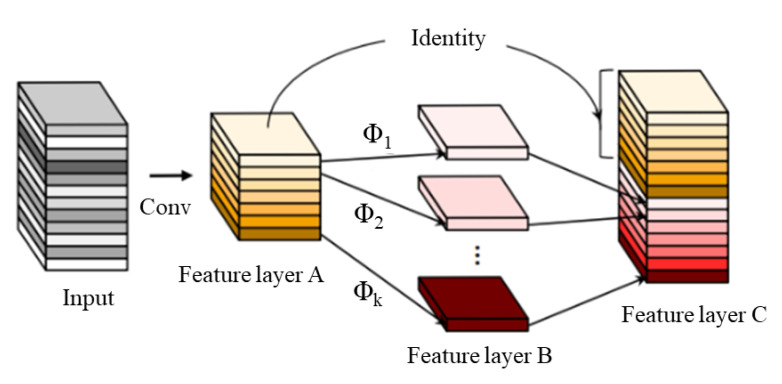
Convolutional process of the Ghost module.

**Figure 7 sensors-23-04335-f007:**
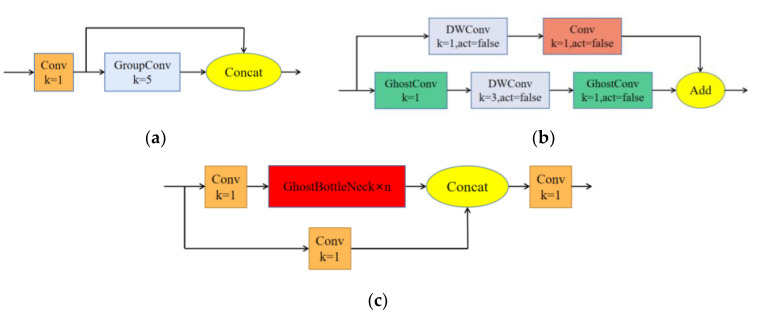
Ghost series modules. (**a**) GhostConv, (**b**) GhostBottleneck, and (**c**) C3Ghost.

**Figure 8 sensors-23-04335-f008:**
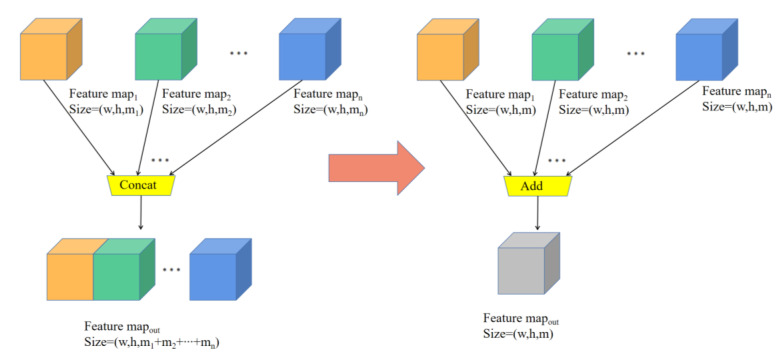
Replace the Concat-based method with the Add-based method.

**Figure 9 sensors-23-04335-f009:**
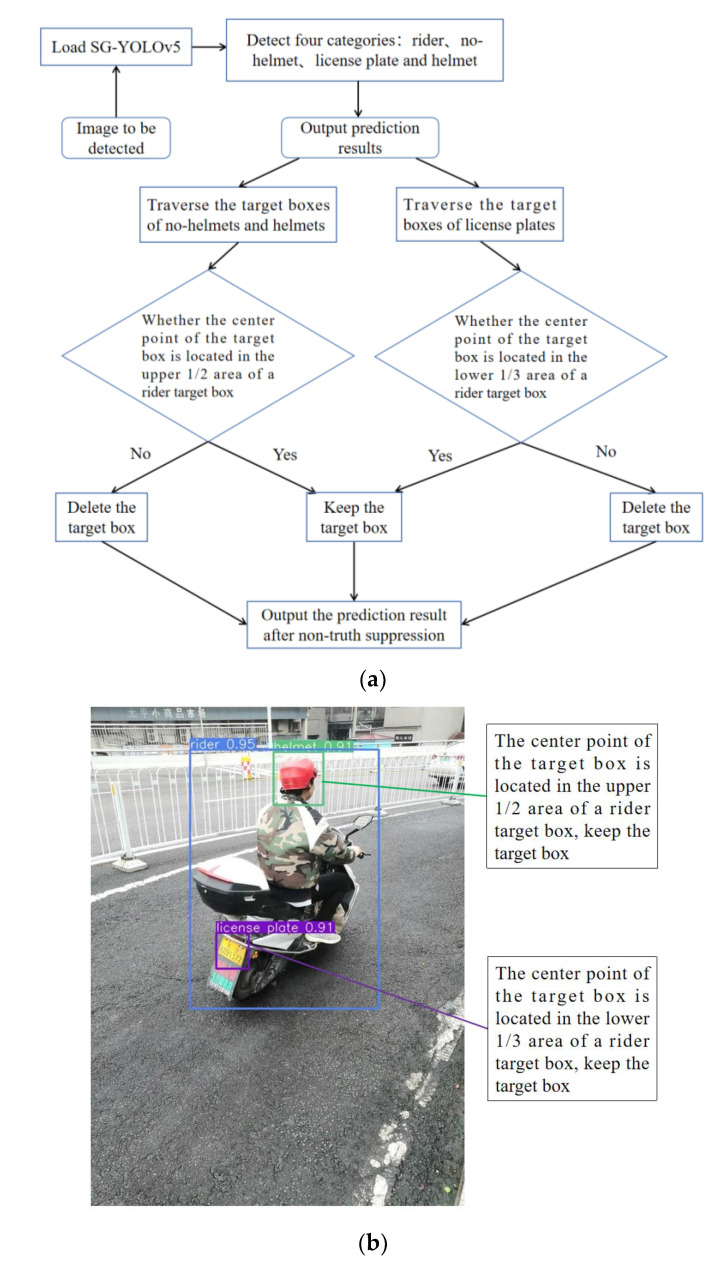
Non-truth suppression method. (**a**) The process of non-truth suppression. (**b**) An example of non-truth suppression.

**Figure 10 sensors-23-04335-f010:**
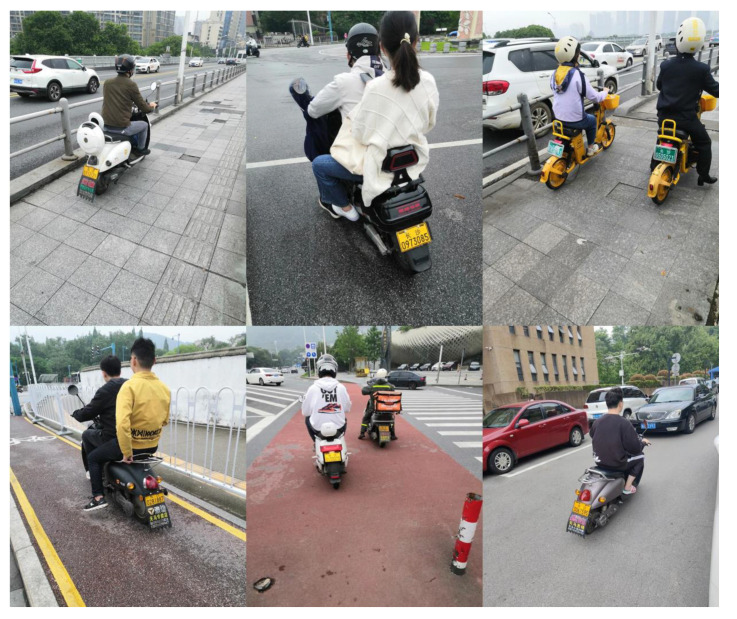
Partial samples of dataset.

**Figure 11 sensors-23-04335-f011:**
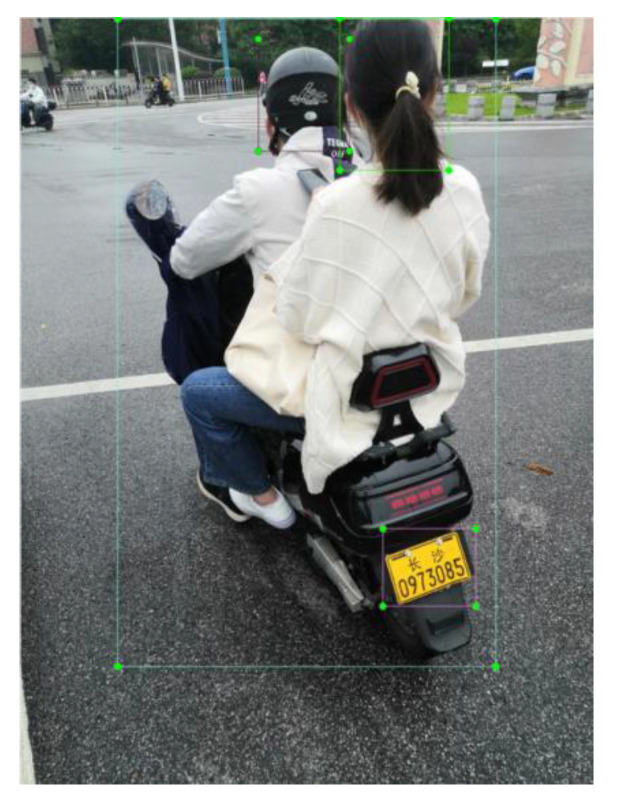
LabelImg software annotation diagram.

**Figure 12 sensors-23-04335-f012:**
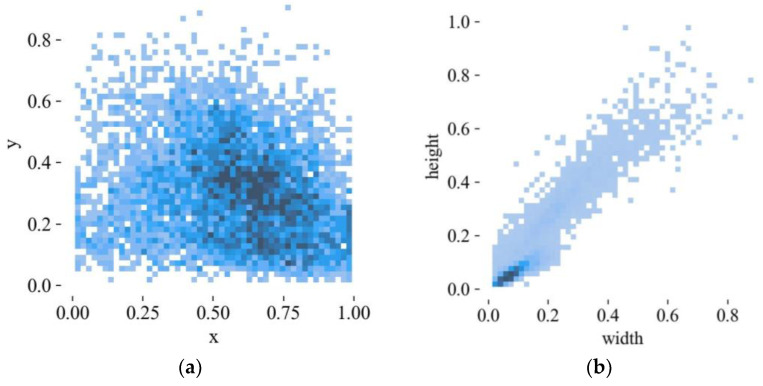
Dataset analysis results. (**a**) Location distribution of dataset target center point. (**b**) Distribution of dataset target size.

**Figure 13 sensors-23-04335-f013:**
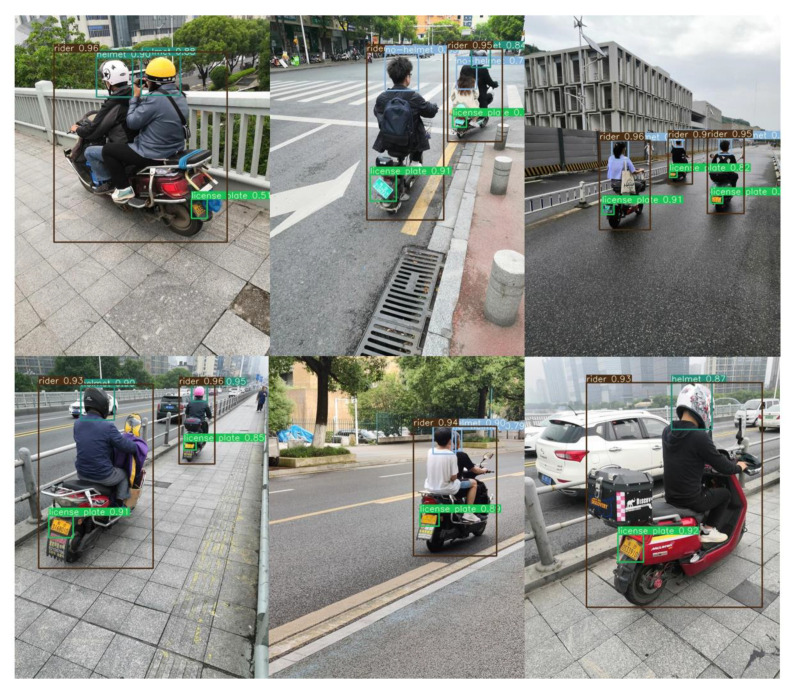
Detection results of SG-YOLOv5.

**Figure 14 sensors-23-04335-f014:**
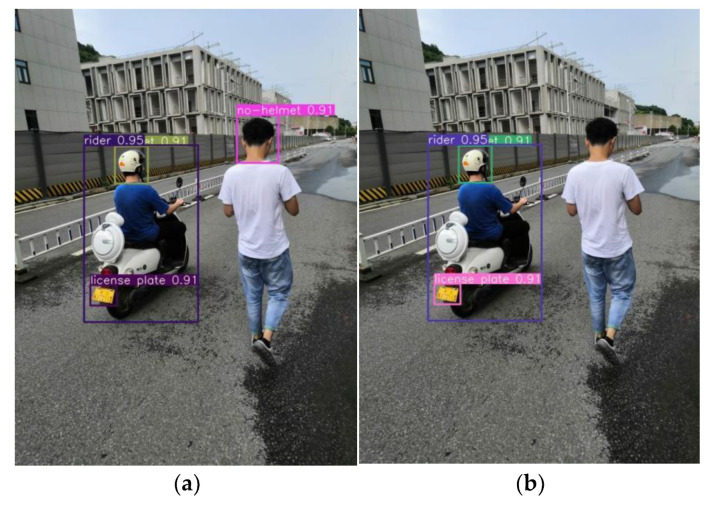
Non-truth suppression experiment. (**a**) No non-truth suppression method. (**b**) Using non-truth suppression method. (**c**) No non-truth suppression method. (**d**) Using non-truth suppression method.

**Figure 15 sensors-23-04335-f015:**
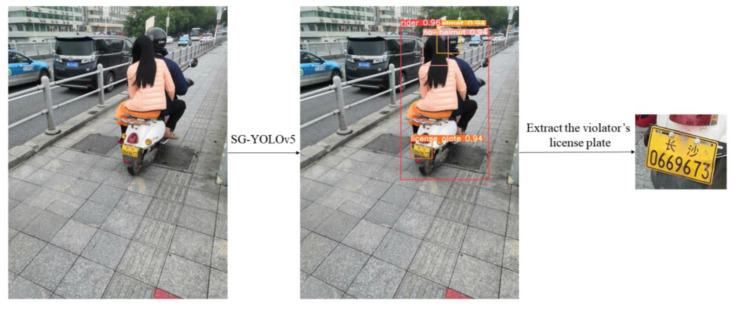
Example of the extraction process for a violator’s license plate.

**Table 1 sensors-23-04335-t001:** SG-YOLOv5 network structure.

Number	From	Params	Module	Arguments
0	−1	3520	Focus	[3, 32, 3]
1	−1	2880	Shuffle_Block	[32, 64, 2]
2	−1	9656	C3Ghost	[64, 64]
3	−1	5280	Shuffle_Block	[64, 64, 2]
4	−1	28,968	C3Ghost	[64, 64]
5	−1	9856	Shuffle_Block	[64, 128, 2]
6	−1	110,160	C3Ghost	[128, 128]
7	−1	36,096	Shuffle_Block	[128, 256, 2]
8	−1	143,072	C3Ghost	[256, 256]
9	−1	18,240	GhostConv	[256, 128, 1, 1]
10	−1	0	Upsample	[None, 2, ‘nearest’]
11	[−1, 6]	0	ADD	[1]
12	−1	33,280	C3Ghost	[128, 128]
13	−1	5024	GhostConv	[128, 64, 1, 1]
14	−1	0	Upsample	[None, 2, ‘nearest’]
15	[−1, 4]	0	ADD	[1]
16	−1	8448	C3Ghost	[64, 64]
17	−1	19,360	GhostConv	[64, 64, 3, 2]
18	[−1, 13]	0	ADD	[1]
19	−1	25,088	C3Ghost	[64, 128]
20	−1	75,584	GhostConv	[128, 128, 3, 2]
21	[−1, 9]	0	ADD	[1]
22	−1	99,328	C3Ghost	[128, 256]
23	[16, 19, 22]	12,177	Detect	-

**Table 2 sensors-23-04335-t002:** Experimental environment configuration.

Parameter	Configuration
CPU	AMD Ryzen 9 5950X 16-Core Processor
GPU	NVIDIA GeForce RTX 3080 Ti
System environment	Windows 10
Acceleration environment	CUDA11.0
Language	Python3.8

**Table 3 sensors-23-04335-t003:** Model comparison during training process.

Epochs		300~349	350~399	400~449	450~499
mAP0.5	YOLOv5s	96.4%	96.4%	96.7%	96.4%
	SG-YOLOv5	96.8%	97.0%	97.3%	97.2%
mAP0.5:0.95	YOLOv5s	67.4%	67.5%	67.6%	67.5%
	SG-YOLOv5	66.3%	66.6%	66.7%	66.9%
precision	YOLOv5s	95.4%	95.6%	95.9%	95.7%
	SG-YOLOv5	95.3%	94.4%	95.4%	95.0%
recall	YOLOv5s	93.3%	93.0%	93.4%	92.7%
	SG-YOLOv5	91.7%	93.1%	92.7%	92.9%

**Table 4 sensors-23-04335-t004:** Comparative experiment.

Model	AP				mAP0.5	FPS	Parameters/10^6^	GFLOPs	Model Size/MB
	Rider	No-Helmet	License Plate	Helmet					
YOLOv3	95.8%	86.6%	95.6%	92.6%	92.6%	4.2	61.54	155.1	117
YOLOv3-tiny	94%	81.1%	94.3%	89.6%	89.8%	39.5	8.68	13.0	16.6
YOLOv3-spp	96.2%	88.2%	96.2%	92.9%	93.4%	4.1	62.59	156.0	119
YOLOv4	97.4%	83.8%	96.8%	90.1%	92.0%	20.9	63.95	59.78	244
SSD	95.2%	74.4%	87.8%	83.1%	85.1%	36.1	24.01	61.11	92.1
YOLOv5s	96.2%	89.7%	97%	93.3%	94.0%	24.9	7.07	16.4	13.7
YOLOv5s-MobileNetv3	94.5%	82.2%	96%	88.3%	90.3%	44.2	3.37	5.9	6.67
YOLOv5s-ShuffleNetv2	97.1%	86.3%	97.3%	91.6%	93.1%	72.5	0.85	1.8	1.82
YOLOv5s-GhostNet	96.5%	88%	96.1%	92.9%	93.4%	37.6	3.92	9.7	7.79
YOLOv7	97.7%	94.3%	96.3%	97.5%	94.7%	2.6	36.50	103.2	71.3
SG-YOLOv5	97.8%	89.6%	97%	92.6%	94.2%	66.7	0.65	3.2	1.54

## Data Availability

Data is available at: https://drive.google.com/file/d/1zESCktHH3VQfsaDviks7v9inztDQ1IXQ/view?usp=sharing, accessed on 16 March 2023.

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
