# Peer review of "Fast Helmet and License Plate Detection Based on Lightweight YOLOv5"

_sensors, 2023, doi:10.3390/s23094335_

Round 1

Reviewer 1 Report

It would be valuable to add a comparison with previously published articles that address a similar type of problem. This would help to contextualize the approach taken and demonstrate how it differs or improves upon existing methods.  

Currently, the lack of a publicly available dataset raises questions about the validity and reproducibility of the results. It is recommended that the dataset be made publicly available for further study and to enable others to replicate or build upon the work. It would also be helpful if the authors could indicate if they plan to release the dataset in future revisions of the paper.

It would be useful for the authors to highlight the specifications of the camera used in the study or clarify whether their solution is equally effective for all types of cameras. This would help readers to understand the applicability and limitations of the approach.

The main question addressed by authors is the detection of helmet and bike number plates, however, there are many studies addressing the same question.   The authors have adopted Yolov5 and there is no other contribution except they have created a dataset, in my previous comments i have asked the authors to provide justifications.   In my opinion, the authors should explain why they are using Yolov5 where there is a latest version available Yolov7   There is no comparison between referenced article and the proposed methodology   My major concern is the dataset, if authors properly address this point then i can decide weather to accept this article or not.

Reviewer 2 Report

In this paper, the authors used ShuffleNetv2 and GhostNet models,  Add-based feature fusion method, and scene-based non-truth suppression to YOLOv5 to improve the fast helmet and license detection of electric bikes. In their model, SG-YOLOv5, by combining ShuffleNetv2 and GhostNet as lightweight. They replaced feature pyramid feature fusion with an Add-based method for feature extraction and used scene-based non-truth suppression to remove interference records and improve model performance in the training phase. Further, the number of parameters and FLOPs have reduced and can be deployed in devices with limited memories. This paper shows a good and promising result and has a good structure but some areas need attention for improvement. Here are some suggestions for the authors to improve their paper.

1. The abbreviation of Name that appeared first, it need to give the full name. Please check the whole paper. 

2.  Why is bias ignored? As stated by the authors, “Here, since the bias has little effect on the total ?????, it is ignored in the calculation…….”. Please explain it detailly.

3.   During training on the RHNP dataset, precisions of YOLOv5s are better than that of SG-YOLOv5 for all the Epochs. The authors should explain the implication.

4.   The limitations of their model (SG-YOLOv5) are not stated. Highlighting these would paint a clear picture of performance and future improvement of the new model.

5.   Please, improve on the body structure of the paper- the introduction should be separated from related works and other structures should be clearly stated. Titles and subtitles should also be numbered clearly.

5. Title of figures should convey and reflect a brief description/explanation/meaning of what they are trying to illustrate, e.g., in figure 3, ‘Focus Structure’ doesn’t convey much meaning in respect of the diagram. Figur1 is shown at the below of section name . It is better after mention, then have a figure showing.

6. There is a need for grammar checks across some sentences by a native expert.

7. There are only two papers which are cited in 2022. The authors are suggested to cite more papers which published in/after 2022.  
